# Sunflower Oilcake as a Potential Source for the Development of Edible Membranes

**DOI:** 10.3390/membranes12080789

**Published:** 2022-08-17

**Authors:** Ancuţa Petraru, Sonia Amariei

**Affiliations:** Faculty of Food Engineering, Stefan cel Mare University of Suceava, 720229 Suceava, Romania

**Keywords:** sunflower oilcake, alginate, glycerol, edible films, film properties

## Abstract

Sunflower oilcake flour (SFOC) resulting from the cold extraction of oil is a rich source of valuable bio-components that stimulated the development of novel, biodegradable and edible films. The films were prepared by incorporating different concentration of sunflower oilcakes (0.1–0.5 g). The obtained films were characterized in terms of physical, water-affinity, antimicrobial and morphological properties. The edible-film properties were affected significantly by the presence and the level of SFOC added. The water vapor permeability and water vapor transmission rate improved with the amount of SFOC added. However, the solubility, oxygen and grease barrier were slightly lower than control film. SEM analysis revealed a rougher but continuous structure with the increases in sunflower oilcake. Moreover, the films with different SFOC levels were opaque, thus presenting good protection against UV radiation. Overall, the SFOC can be use as raw material to produce edible films with suitable properties and microbiological stability for food-packaging applications.

## 1. Introduction

Food packaging is very important because it provides nutritional information for consumers and protection against potential damage (physical) and environment contamination (chemical and microbiological factors) [1]. Usually, the materials used for packaging are glass, paper, metals, plastics and polymeric materials. From these, plastics are preferable due to their good material properties (low cost, good tensile strength and good protection against moisture, oxygen, unpleasant odor and microorganisms) [2]. The major disadvantage of plastic is the crucial effect on the environment: non-degradability and non-renewability [3].

Edible and biodegradable films are upcoming alternative packaging materials for reducing plastic waste while improving stability, quality, safety and the variety offered to consumers [4].

Edible packaging materials are obtained from natural polymers such as polysaccharides, lipids, proteins or a combination of these [5]. The continuous development of edible film has been created by various researchers to match conventional plastic films [6]. Polysaccharides (cellulose, starch, pectin, alginates and chitosan) are the most popular natural polymer used in the production of edible films [7]. Alginates are isolated from brown seaweeds and, due to their properties (thickening; stabilizing; film-forming; suspending; resistance to solvents, oil and grease), are competent materials for the development of edible films [8,9]. 

The application of edible films as a packaging material is influenced by their characteristics, such as structural, biological, optical and barrier. The edible films should exhibit a good barrier against scents, vapor, oil and water, as well as oxygen and light degradation (inhibition of lipid oxidation, delaying moisture loss, prevention of discoloration, maintenance of the products appearance during marketing), excellent solubility and antimicrobial properties (improvement of the quality and shelf life of the products) [10]. Sensory properties are also important in the production of packaging. Thus, edible films can be considered for commercialization if they fulfill the most important factor, i.e., edibility. To be eaten as part of food, all ingredients in the edible films should be GRAS (general recognized as safe) and used within the limitations specified by the U.S. Food and Drug Administration (FDA) [11,12]. 

In recent decades, due to the increase in the popularity of sustainability and environment-protection concepts, it is important in industrial production to develop new strategies designed to make the best use of all resources without creating any wastes [13,14]. Oilseeds are grains that, due to their high fat content (>40%), are used primary in most countries as sources for vegetable oils [15]. The oilseed industry generate large amounts of byproducts that are currently underused [16]. Sunflower oilcake (SFOC) is a byproduct that remains after the cold extraction of oil from sunflower seeds [17]. SFOC contains significant amount of residual oil (1–23.6%), proteins (19.93–44.9%), minerals (4.69–8%), fibers (13.07–33.4%) and carbohydrates (15–28.2%) [18]. It can be used in both human and animal diets due to its rich content in bioactive compounds [18]. Possible methods of valorization include: the isolation of high-value compounds and their further utilization in foods; as a substrate for the production of fuels, surfactants, enzymes and antibiotics; and as feeds [19,20,21]. In recent year the utilization of residues and byproducts from the food industry has aroused great interest for the production of edible packaging material [22]. Byproduct valorization allow the reuse of these materials into the supply chain, thus adding more value to foods, also reducing cost and risks regarding their disposal in the environment [13,20,23].

Suput et al. [24,25] investigated the possibility of using whole sunflower oilcake in the development of biopolymer films and the effects of pH, temperature and essential oil on them. The films were firm, smooth, shiny and dark brownish/greenish, with a sunflower fragrance. Water vapor permeability and solubility were uniform but decreased with increases in temperature. The tensile strength and elongation at break were fairly low, but antioxidant properties and increased elongation at break were obtained when adding 0.25–1% parsley and rosemary oils. All of these properties make the films sufficient for application in the food industry. The films had also a good barrier to UV.

In the literature were found studies when other oilcakes were used in the production of films, such as pumpkin, hemp and rapeseed. Popovic [26,27] obtained pumpkin-oilcake-based films with adequate characteristics. With the increase in pumpkin oilcake amount were obtained films with the best tensile strength, elongation at break and solubility properties. Due to the content in antinutritional factors, the proteins were extracted from hempseed and rapeseed oilcake. With 50% glycerol (as plasticizer) and the extracted proteins were obtained high-performance films at pH 12 [28]. The realization of only-rapeseed-proteins-based films cannot be done because the films presented bad mechanical and antimicrobial properties; thus, gelatin, chitosan and agarose must be used [29,30].

In order to propose an added-value outlet for the SFOC, the present work was aimed at investigating the potential of SFOC in the preparation of new edible packaging materials. Thus, films with different amounts of SFOC were examined to determine the effect of the oilcake on the water-affinity, mechanical, optical, barrier and structural properties of the SFOC-based films.

## 2. Materials and Methods

### 2.1. Materials

Sunflower oilcake (SFOC) was collected from a local factory in Suceava, Romania. The material was ground to a fraction of less than 180 µm and stored at room temperature until further use. 

All chemicals used in this paper were of analytical grade and were purchased from Carl Roth (Karlsruhe, Germany). 

### 2.2. Film Development

Films were developed by a wet cast method using sodium alginate, sunflower oilcake and glycerol as plasticizer. The sodium alginate was dissolved in distilled water (1 g in 100 mL) at 50 °C for 1 h using a constant-temperature magnetic stirrer DLAB MS-H-PRO^+^ (Beijing, China). After complete dissolution, 0.5 g of glycerol and different proportions (0.1–0.5 g) of SFOC were added in the solution. Film formulations are presented in Table 1.

The film solutions were poured into Petri dishes and dried at 50 °C in an air oven for 48 h. The obtained films were kept in sealed envelopes at 20 °C and 50% relative humidity (RH) before further tests.

### 2.3. SFOC Characterization

Edible films are an integral part of the edible portion of food products, so they should follow the required regulation for food ingredients. Therefore, for every ingredient introduced in the film, its safety must be clearly demonstrated. The safety of the sunflower oilcake was demonstrated with the following analysis: water activity, spectroscopic methods and ELISA method. 

Water activity index (a_w_) was measured using an AquaLab 4TE water activity meter (Meter Group, Pullman, WA, USA) [31].

The ELISA (enzyme-linked immunosorbent assay) method was determined using kits provided by ProGnosis Biotech S.A. (Larissa, Greece) [32]. The samples were analyzed for the content of zearalenone, ochratoxin A, aflatoxin B1 and deoxynivalenol. 

The mineral elements were determined with coupled mass spectrometer (Agilent Technologies 7500 Series, Santa Clara, CA, USA) in order to highlight the possible contamination of the sample with heavy metals.

### 2.4. Film Characterization

#### 2.4.1. Affinity to Water

Water content (WC) was determined by gravimetric method [27]. A piece of each film (3 cm × 3 cm) was dried at 110 °C for 24 h in a laboratory oven, ZRD-A5055 (Zhicheng Analysis Instruments, Shanghai, China). WC was calculated using Equation (1):(1)$$\mathrm{WC}=\frac{W_{0}-W_{1}}{W_{0}}\times 100$$
where W_0_ is the mass of the film before drying (g), and W_1_ is the mass of the film after drying (g).

The water activity of the films was determined according to the methods described previously.

The water solubility (WS) of the control and SFOC films was determined by immersing 3 cm × 3 cm specimens into 30 mL of water. The solution was gently stirred at room temperature for 8 h. The remaining films were filtered and dried at 100 °C for 24 h. The results were expressed as a percentage of the films before and after solubilization [33,34,35].

#### 2.4.2. Barrier Properties

The water vapor permeability (WVP) of films was determined using a gravimetric method (ASTM E96-96M, 2016). The films were sealed on plastic Petri dishes filled with calcium chloride (RH = 0%) up to 1 cm from the film underside. The membranes were placed inside a desiccator containing saturated sodium chloride solution (RH = 75%). Due to the fact that the RH inside the dishes was lower than that from outside, the WVP was recorded using the weight gain of the dishes [36]. The dishes were weighed every eight hours for 48 h. Five weight measurement were made at 0, 8, 24, 32 and 48 h. The change in weight was recorded as a function of time; then the slope of each line was calculated by linear regression.

The water vapor transmission rate (WVTR) was calculated as the slope (g/h) divided by the film area (m^2^) [37]. Results were calculated by Equations (2)–(4):(2)$$\mathrm{WVTR}=\frac{w}{A\times t}$$
(3)$$\mathrm{WVP}=\frac{\mathrm{WVTR}\times x}{∆p}$$
(4)$$∆p=S\times \left(R_{1}-R_{2}\right)$$
where w/t is the weight gain of the dishes in time (g/h), x is the average film thickness (mm), A is the area of the exposed film (m^2^), Δp represents the partial water vapor difference across the two sides of the films (kPa), S is the saturated vapor pressure at 25 °C (3166 kPa), R_1_ is the relative humidity in the desiccator (0.75), and R_2_ is the relative humidity inside the dishes (0) [38,39,40].

The oxygen permeability (OxyP) of the films was determined by the analysis of the peroxide value. The membranes were cut into circles (diameter 3 cm) and sealed on top of Erlenmeyer flasks filled with 3 g sunflower oil. Then, the flasks were stored at 50 °C for 10 days [41]. The peroxide value (PV) was determined according to standard AOAC 965.33, with some modifications. The oil was treated with 10 mL chloroform, 15 mL glacial acetic acid and 1 mL potassium iodide. The samples were kept in the dark for 5 min; after that, 75 mL distilled water and 1 mL of starch were added. After vigorous stirring, the samples were titrated with 0.01 N sodium thiosulphate solution until the blue color disappeared.

Oil permeability (OP) was determined using the method described by Cao et al. (2020) [42], with some modifications. The films were cut in circles with a diameter of 3 cm and sealed on top of tubes containing 20 mL of sunflower oil. The tubes were placed upside down for 5 days on pre-weighed filter paper. PO was calculated by using the equation presented below:(5)$$\mathrm{OP}=\frac{\left(∆m\;\times x\right)}{S\times t}$$
where Δm is the changes in filter paper mass (g), x is the thickness of the tested film (mm), S is the surface of the film (m^2^), and t is the testing time (days).

#### 2.4.3. Thickness, Density, Tensile Strength and Hardness

Film thickness was determined using a digital micrometer Mitutoyo Absolute (Kawasaki, Japan) with an accuracy of 1 µm. The results were obtained after ten readings on different areas of the film’s surface [31].

Density (p, g × cm^−3^) was calculates from the film dimensions according to the following Formula (6) [43]:(6)$$p=\frac{m}{A\times x}$$
where m is the mass (g), x is the thickness (cm), and A (m^2^) is the area of the films.

Tensile strength (TS) and hardness were measured using a Perten TVT 6700 texture analyzer with a 5 mm deformation and a 5 mm cylinder probe. The pre-speed, test speed and post speed were 2, 1 and 10 mm/s. The results were expressed in Newtons. The TS was calculated according to Equation (7):(7)$$\mathrm{TS}=\frac{F_{\max}}{\left(x\times W\right)}$$
where F_max_ is the maximum tensile force at rupture (N), x is the thickness (mm), and W (mm) is the width of the films.

#### 2.4.4. Optical Properties

The color analysis of the samples was performed with a CR-400 colorimeter (Konica Minolta, Tokyo, Japan) using the CIELAB scale, wherein the L* value expresses lightness (0 for black and 100 for white), the a* value expresses the degree of redness (if the value is positive) or greenness (if the value is negative), and the b* value expresses the degree of yellowness and blueness (if the value is positive or negative, respectively) [44]. The color difference (ΔE) was determined using Equation (8), described below:(8)$$∆E=\sqrt{{\left(∆L^{*}\right)}^{2}+{\left(∆a^{*}\right)}^{2}+{\left(∆b^{*}\right)}^{2}}$$
where ΔL*, Δa* and Δb* represents the color parameters differential between the samples and the standard white plate used as background (L* = 94.27, a* = −5.52, b* = 9.19).

For each sample were taken five readings (one in the center and four around the surface areas). Color measurements were taken in triplicates [45]. 

The films were cut into strips (1 cm × 3 cm) and placed inside a spectrophotometer test cell. All of the measurements were performed using air as the blank reference [28].

The ultraviolet and light spectrum was obtained using a Shimadzu 1800 UV spectrophotometer by exposing the films to light at wavelengths ranging from 200 to 800 nm [24]. 

The opacity and transparency of the films were calculated using the Equations (9) and (10) below [41,46]:(9)$$\mathrm{Opacity}=\frac{{\mathrm{Abs}}_{600}}{x}$$
(10)$$\mathrm{Transparency}=\frac{\log T_{600}}{X}$$
where Abs_600_ and T_600_ represent the absorbance (UA) and transmittance (%) at 600 nm, and t represents the thickness of the films (mm).

#### 2.4.5. FT-IR

The membranes, pure alginate, glycerol and SFOC powder were analyzed using FT-IR spectroscopy. The spectra were recorded within the range of 400 cm^−1^ to 4000 cm^−1^ using a Nicolet iS20 spectrometer (Thermo Scientific, Karlsruhe, Germany), equipped with an attenuated total reflectance accessory and a diamond crystal. Spectra were collected at 4 cm^−1^ resolution and 32 scans. The obtained spectra were processed with OMNIC software [37,47].

#### 2.4.6. Scanning Electron Microscopy (SEM)

The surface and cross-section of the obtained films were analyzed by SEM (Tescan Vega II LMU, Tescan Orsay Holding, Brno, Czech Republic). The films were cut into small pieces and were fixed on double-sided adhesive carbon bands. The images were analyzed at an acceleration of 30 kV and were collected at 1 kx and 700× magnifications. 

#### 2.4.7. Antimicrobial Analysis

For the microbiological assessment of the samples were used compact dry-type plates with lyophilized culture media. The samples (1 g) were dissolved in 9 mL of saline solution. Then, 1 mL of this solution was dispersed on the culture media [34]. The films were tested for the total number of germs, *Escherichia coli*, *Staphylococcus aureus*, *Listeria*, coliforms, *enterococcus*, *Bacillus cereus*, yeasts, molds, *Enterobacteriaceae* and *Salmonella*. For the first analysis, the plates were kept in a hot air oven at 35 °C for 48 h (AOAC 010404). For the following five previously cited microorganisms, the plates were kept at the same temperature but for 24 h (AOAC 110402, 081001, ISO 11290-2:2017, 110401, 11190). For *Bacillus* cereus, yeast and molds the plates were kept at 30 °C for 2 (MicroVal 2019-LR87) and 3 days (AOAC 100401), respectively. For *Enterobacteriaceae* and *Salmonella* the plates were kept for 24 h at 37 °C (AOAC 012001) and 42 °C (ISO 6579-1:2017) respectively.

### 2.5. Statistical Analysis

Results were presented as mean ± standard deviation. The WC, WS, WVP, WVTR, OxyP, OP and density determinations were performed in triplicates. The results were processed using XLSTAT (trial version). The difference between the films were evaluated by ANOVA, using a Turkey test at a 95% confidence level. A principal component analysis (PCA) was applied to observe the relationships between the water-affinity and the optical and barrier properties of the obtained membranes.

## 3. Results

### 3.1. Sunflower Oilcake Characterization

The chemical composition of SFOC flour obtained from a previous study was [48]: 8.75 ± 0.10% moisture, 20.15 ± 1.57% proteins, 15.77 ± 0.45% lipids, 4.56 ± 0.11% ash, 31.88 ± 0.79% crude fiber and 18.89 ± 0.23% carbohydrates.

The value obtained for water activity was low (0.40 ± 0.01), smaller than 0.6, which does not allow the growth of molds, yeast and bacteria [49]. 

The results for the mycotoxins studied were within the allowed legal limit established by the European Union and presented in Table 2.

Mineral analysis by ICP–MS showed the absence of heavy metals, such as lead, mercury and cadmium [48].

### 3.2. Water Affinity

The properties regarding the affinity to water of the obtained films are presented in Table 3. The water-content values varied from 13.07 to 19.45%, showing a significant difference (*p* > 0.05) between the control and the sunflower oilcake films. The water content decreased with the amount of SFOC introduced. It can be assumed that the increase in oilcake flour, which is a lipophilic substance, does not allow water incorporation. Moreover, Bahmid et al. (2021) [50] declared that moisture may depend on thickness and surface roughness (particle dimensions, their amount and evaporation conditions), because the particles can make water penetration more difficult. This conclusion was also observed in our results, so the films containing ground SFOC stimulated slower water absorption. The control film had higher moisture due to the abundant presence of the hydrophilic group in sodium alginate [37]. 

a_w_ is an important factor that affects the quality and shelf life of food products during processing and storage [51]. The parameter is determined based on the moisture levels and the interactions between water molecules and other ingredients [52]. All of the samples presented good a_w_ values (between 0.29 and 0.40) and thus were considerate not susceptible to microorganism growth [53]. The values increased with the amount of oilcake added; this may be due to the increasing content in components (proteins, fiber and carbohydrates) that retain more water [54].

The solubility is an important requirement for the films because through this its depends the potential use of the edible films (e.g., the encapsulation of food additive or the maintenance of product integrity) an important requirement is the solubility. WS reflects the water resistance of the films. The solubility obtained for the films was high. When sunflower oilcake was added a significant (*p* > 95%) decrease in solubility was observed; thus, the lowest value was found for SFOC5. The decrease may be related to the increase in solid components, especially fats and fibers. Other reasons can be the differences in thickness and the inhomogeneous structure of the membranes [54].

### 3.3. Barrier Properties

#### 3.3.1. Water Vapor Permeability (WVP) and Water Vapor Transmission Rate (WVTR)

In Table 4 are presented the barrier properties of the films. WVP is an indicator of the membrane’s capacity to prevent moisture transfer. A large value negatively influences the quality and life of foods [55]. This factor is influenced by thickness, components, humidity and water activity [37].

WVP was significantly (*p* < 0.05) affected by the amount of SFOC added; the values decreased from 1.98 × 10^−4^ to 1.13 × 10^−4^ g × mm/KPa × h × m^2^. The values obtained indicated an improvement in the vapor barrier properties of the films. The decrease may be due to the higher concentration of oilcake, which makes the films thicker and denser. The values found were lower than those reported by Suput et al. (2018) [24] and Hromis et al. (2019) for films based on whole sunflower and pumpkin oilcakes. As expected, the lowest WVTR values was recorded for the sample with the highest sunflower oilcake addition.

The low moisture barrier of the control sample is caused by the behavior of alginate, which accelerates the water uptake, the permeability and transmission rate of the vapor [56]. 

#### 3.3.2. Oxygen Permeability (OxyP)

The peroxide value (PV) obtained for the uncovered oil was 7.67 meq O_2_/Kg, which was higher that the PV values obtained for the oil covered with the obtained membranes. The results indicates that SFOC could block oxygen transmission. The PV value of the oil covered with the developed films ranged from 2.24 to 4.83 meq O_2_/Kg. The increase may be related to the high fat content of the oilcake.

The results indicates that all of the membranes were able to form a good barrier against atmospheric oxygen, protecting the foods to be packaged from unwanted oxidation reactions.

#### 3.3.3. Oil Permeability (OP)

The OP analysis was done to investigate the practical application of the obtained membranes in fatty food packaging. As shown in Table 4, the OP of the control film was 0.027 g × mm × m^−2^ × day^−1^. After adding SFOC, the values increased from 0.017 to 0.034 g × mm × m^−2^ × day^−1^. No significant difference (*p* > 95%) were found between the control sample and the membranes with 0.1 g and 0.2 g SFOC. On the contrary, significant differences (*p* < 0.05) were found when the addition of flour increased (0.4–0.5 g). The results obtained were much lower than those obtained by Cao et al. [42] in edible films obtained with cassia gum reinforced with carboxylate cellulose nano crystal whisker (0.5, 0.1, 0.064 and 0.067 g × mm × m^−2^ × day^−1^). Therefore, SFOC-based films are promising for the packaging of oil-rich foods.

Moreover, on the filter paper, stains were not observed; this confirms the excellent barrier properties of the obtained membranes against grease. The alginate-based films are highly hygroscopic; thus, the oil, which is a lipophilic substance, does not dissolve in the alginate films [9,56]. 

### 3.4. Thickness, Density, Tensile Strength and Hardness

The thickness values varied between 0.029 to 0.044 mm, showing a significant difference (*p* < 0.05) between the SFOC and control films. The thickness increased with the increasing of the SFOC amount, and the differences were significant (confidence level 95%). This increase may be due to: the increase in solid contents (SFOC), the various differences in structure and chemical composition (high fiber content of SFOC) and the hydrophobicity character of the film constituents. The density of the film decreased from 1.45 g/cm^3^ to 1.04 g/cm^3^ with the addition of SFOC due to the increase in thickness.

The results for the mechanical characteristics are shown in Table 5. In comparison with other studies, the tensile strength differs (1.11 MPa [37], 1.23 MPa [57]); this may be due to the method of the preparation of the films (such as dry matter content, composition, mixing time, drying parameters and thickness) [56].

When increasing the SFOC content, a significant tendency of tensile strength and hardness to decrease was observed in the films.

### 3.5. Optical Properties

#### 3.5.1. Color Analysis

The optical properties of the food packaging material are important because the acceptance of the products by consumers depends on them. In Table 6 are presented the color values for the control and the SFOC films.

The L*, a* and b* values are in the range of 73.10 to 92.97, −2.10 to −5.53 and 10.3 to 23.46, respectively. 

SFOC films showed a darker color than the control film (L* values decrease); the differences were significant different at *p* < 0.05. The amount of SFOC significantly influenced (95% confidence level) the lightness of the samples (Figure 1).

Regarding the values obtained for the chromatic coordinates a* and b*, significant differences (*p* < 0.05) were observed between the control and the SFOC films. When increasing the amount of SFOC, a* values decreased while b * values increased significantly (*p* > 95%). Negative a* and positive b* coordinates imply a predominant greenish/yellowish coloration in the films. The a* value may be related to the predominance of SFOC or may be due to the chemical composition, in particular the presence of pigments in SFOC flour. 

The total color differences were determined to see if the addition of oilcake influenced the overall membrane coloration. The results increased significantly from 1.74 to 25.77, with a significant difference (*p* < 0.05), demonstrating that the color difference can be distinguishable with the naked eye.

#### 3.5.2. UV–VIS Spectra

The sensory and nutritional qualities of food products can be altered by exposure to light. The requirements for the spectral properties of packaging materials regarding UV–VIS are the low transmittance for UV radiation (thus increasing the lifespan of the packaged foodstuff) and high transparency in the visible region (to provide consumers with visual control). The protection against UV radiation is the most important requirement because radiation can cause deterioration in the packaging material [58].

Compared to the control sample, the SFOC films presented good light absorption between 200 nm and 400 nm. Considering the fact that this is in the UV spectra range, the samples have the ability to protect products against UV radiation. As shown in Figure 2, the pattern of light absorption in the UV region was: SFOC5 > SFOC4 > SFOC3 > SFOC2 > SFOC1 > control. 

In the visible field (400–800 nm), all samples presented low absorption values, thus providing high visual access. Moreover, the absorption of the SFOC membranes in the visible field were higher than the absorption for the control sample. This is due to the darkening of the membranes as the SFOC was added.

When SFOC was added, the transmission values in the UV decreased significantly (*p* < 0.05). In our study, the lowest transmission, as well as the best UV barrier, was found in the sample with 0.5 g addition.

Transparency is an important property for marketing, because a transparent material is usually more attractive for consumers [59]. The transparency and opacity values are summarized in Table 5. It can be seen that the addition of oilcake had a statistically significant (*p* < 0.05) impact on these properties. Higher addition levels resulted in lower transparency values and respectively higher opacity values. The results obtained were in accordance with the transmittance values, so it can be said that the opaquer films could be used as food protectants against light permeability. 

### 3.6. FT-IR

The location and intensity of the characteristic absorption peaks for the sole ingredients (pure alginate, glycerol and SFOC powder), as well as for the control and SFOC membranes, are shown in Figure 3. The FTIR spectra of all of the obtained membranes presented four absorption wave numbers in three different spectra zones, namely 3500–3200 cm^−1^, 3000–2800 cm^−1^ and 1000–1030 cm^−1^, which can be associated to the bond stretching of O–H, C–H (symmetric and asymmetric) and C–O–C groups. Moreover, these are the characteristic broad bands present in the alginate structure [60]. 

Some of the peaks shifted to a lower (2927.14 cm^−1^ to 2920.29 cm^−1^) and/or higher (from 323.50 cm^−1^ to 328.52 cm^−1^ and from 1023.46 cm^−1^ to 1024.51 cm^−1^) intensity with the increase of SFOC, which was indicative of the interactions between the sodium alginate and the oilcake flour. Moreover, in comparison with the control samples in the SFOC films, there was found an additional peak at 1743–1744 cm^−1^, possibly attributed to the carbonyl ester group present in lipidic molecules [61,62]. The films exhibited a pronounced absorbance band between 1600 cm^−1^ and 1300 cm^−1^, which corresponds to the asymmetrical and symmetrical stretching of the COO bond [37].

The spectrum of pure glycerol exhibited characteristic absorption bands at 850.14, 921.60, 992.58, 1028.07 and 1107.46 cm^−1^, maybe corresponding to the vibration of C-H, C-O linkage and stretching.

The spectra of SFOC showed an absorption band at 1634.51 cm^−1^ and 1540.32 cm^−1^, corresponding to the amide I and II region, respectively [63]. Other absorption peaks presented in the spectra are: 1743.46, 1237.75 and 1033.97 cm^−1^, which may correspond to the COO (for unconjugated cellulose) and C-O stretching [64]. Peaks between 2900 and 3000 cm^−1^ may correspond to the asymmetric stretching of CH_2_ and CH characteristic for hemicellulose and cellulose [65].

### 3.7. Apparence and Morphologhy

With the naked eye, the control film was colorless and transparent, while the SFOC-based films were brownish, very shiny and had a slight sunflower flagrance. All films presented a homogeneous structure, are easy to handle and are resistant when manipulated by hand. These appearance characteristics were observed also by Lazic et al. (2020) when they developed SFOC-based films [16].

SEM analysis was conducted to observe the arrangements of the film components and the morphological differences of the developed membranes when the SFOC was added [66].

In Figure 4 is presented the SEM micrograph of the control and SFOC composite films. The control sample presented a smooth and uniform structure. On the contrary, Luo et al. (2019) [37] observed several cracks and bulges in plain sodium alginate films. With the addition of sunflower oilcake, the films showed rougher, but still continuous, structures. The roughness increased with the increasing amount of SFOC added, due to the increase in fibrous particles. 

Moreover, no pore or ruptures were found on the cross-section structure of all membranes, indicating that the films were dense and continuous.

A compact structure may contribute to the obtention of a low WVP value and good mechanical properties [37]. 

### 3.8. Microbiological Stability

Microbial contamination is the main cause of the spoilage and unacceptability of different food products [67]. According to the results presented in Table 7, the films with SFOC showed high microbiological stability. No coliform, *Enterobacteriaceae*, *E. coli*, *Salmonella*, *Staphylococcus aureus* or *Listeria* developed on the culture media. Regarding the total count, the highest values were found for SFOC5 due to the high content of vegetable material. All of the values obtained were within the permissive limit set by food safety and standard regulations (FSSAI), European regulations, and the Food and Drug Administration (FDA). In SFOC 5 were also observed more microorganism (*Enterococcus,* yeast, molds and *Bacillus cereus)* than in the other sample, by precisely 1 cfu due to the increased content of SFOC, but the values were far below the maximum allowed limit set by FSSAI and FDA.

Moreover, we observed that the plain sodium film also exhibited antimicrobial ability against nine microorganisms. The results obtained for *E. coli* and *S. aureus* were in accordance with those obtained in a previous cited study [37]. 

The microbial stability of the films is also due to the presence of sodium alginate, which gives general protection that increases the resistance against microbial agents [68,69].

**Table 7 membranes-12-00789-t007:** Microbial analysis of the control and SFOC membranes.

Microorganism	Control, cfu	SFOC1, cfu	SFOC2, cfu	SFOC3, cfu	SFOC4, cfu	SFOC5, cfu	Limits
Total count	1	2	1	4	4	15	<50 cfu ^1^ <100 cfu ^2^
*Enterococcus*	Absent	Absent	Absent	Absent	Absent	1	<150 cfu ^4^
*Coliforms*	Absent	Absent	Absent	Absent	Absent	Absent	<10 cfu ^1^ Absent ^2^ <50 cfu ^3^
*Enterobacteriaceae*	Absent	Absent	Absent	Absent	Absent	Absent	Absent^1^
Yeast and molds	Absent	Absent	Absent	Absent	Absent	1	<100 cfu ^1^ Absent ^2^ <10^2^ cfu ^3^
*Bacillus cereus*	Absent	Absent	Absent	Absent	Absent	1	10^2^–10^3 5^
*E. coli*	Absent	Absent	Absent	Absent	Absent	Absent	Absent ^1,2,3^
*Salmonella*	Absent	Absent	Absent	Absent	Absent	Absent	Absent ^1,2,3^
*Staphylococcus aureus*	Absent	Absent	Absent	Absent	Absent	Absent	Absent ^1,2^
*Listeria*	Absent	Absent	Absent	Absent	Absent	Absent	Absent ^1,2^

^1^ microbiological limit for fruit, vegetables and seeds according to food safety and standard regulations (FSSAI); ^2^ microbiological limits for packaging materials according to food safety and standard regulations (FSSAI); ^3^ microbiological limits for fruit, vegetables and seeds according to FDA circular No. 2013-010 (revised guidelines for the assessment of microbiological quality of processed food); ^4^ microbiological limits for vegetables found by Julien et al. (2017) [70]; ^5^ microbiological limits found by Yu et al. (2019) [71].

### 3.9. Statistical Analysis

The relationship between the water-affinity properties, barrier characteristics, optical properties, density and thickness are presented in Figure 5. The two principal components explained 96.79% of the total variance (PC1 = 80.66% and PC2 = 16.12%). The PC1 was associated with the optical properties (L*, a*, b*, ΔE*, opacity, transparency and transmittance), water-affinity properties (a_w_, moisture, time of solubility, solubility), barrier properties (WVTR and WVP), density and thickness. On the other hand, only OP and PV were associated with PC2. Regarding the samples, good relationships were observed between SFOC1 and SFOC2 and between SFOC4 and SFOC5.

High positive correlations were found between the a*, b*, ΔE*, opacity, time of solubility and thickness parameters. Other positive correlation were found between moisture, WVTR and density, between transmittance and transparency and between L* and solubility. Negative correlations were found between the optical properties, water-affinity properties and barrier properties.

## 4. Conclusions

Sunflower oilcake obtained after the cold extraction of oil was investigated as a potential source for edible films. The use of sunflower oilcake has led to an increase in film properties and their nutritional value. 

The results showed that, with the addition, the thickness, water activity, time of solubility, oxygen and oil permeability increased, while the moisture, solubility, water vapor permeability decreased. The SFOC composite films exhibited high absorption of UV radiation, thus protecting foodstuffs against photochemical reactions. On the other hand, the absorption in the visible field decreased, indicating a decrease in film transparency. Regarding the structure, the membranes were homogeneous and compact, without pores and cracks. Moreover, the films showed microbial stability against six tested microorganisms, which make them safe to be consumed directly with the products chosen to be packaged. The abovementioned properties make the membranes suitable for the packaging of a wide range of foods, including those susceptible to oxidative changes. Their good solubility also makes them suitable for the packaging of powdery products that need to be dissolved in hot water. Another possible application can be found in the packaging of sliced products (meat, cheese), due to the growing interest of consumers in smaller portions. 

## Figures and Tables

**Figure 1 membranes-12-00789-f001:**
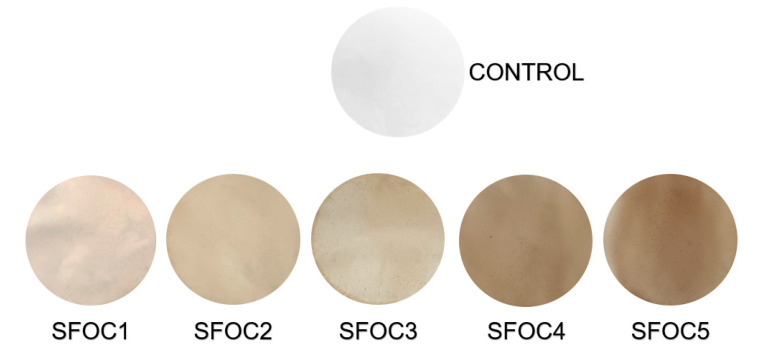
Color variation between the samples with SFOC addition and the control sample.

**Figure 2 membranes-12-00789-f002:**
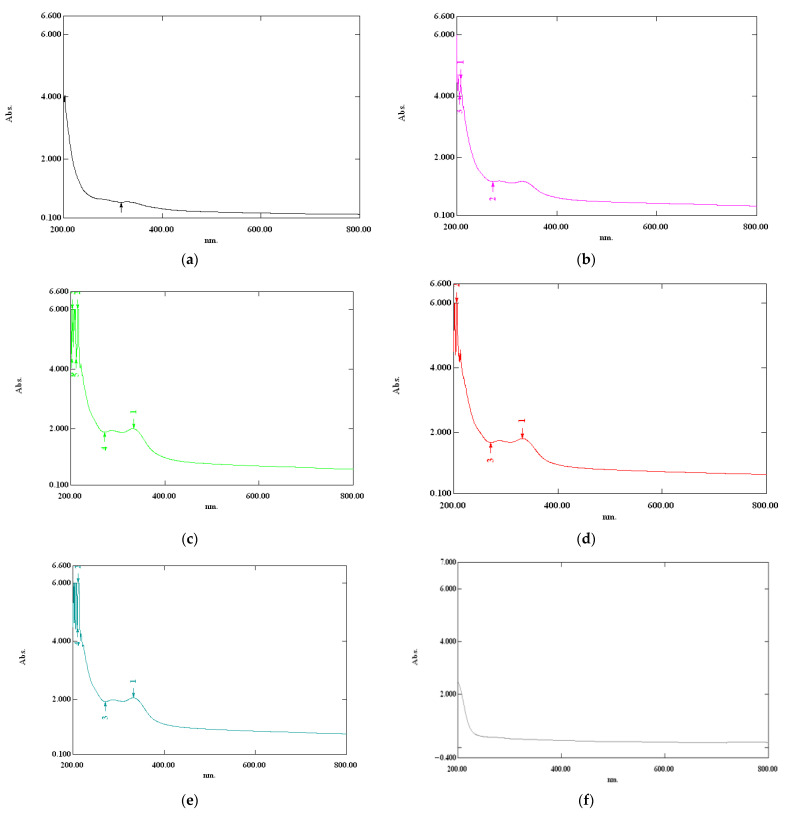
Absorption spectra between 200 and 800 nm of the membrane with SFOC and control samples: (**a**–**e**) are membranes with 0.1, 0.2, 0.3, 0.4 and 0.5 g SFOC, respectively; (**f**): control sample.

**Figure 3 membranes-12-00789-f003:**
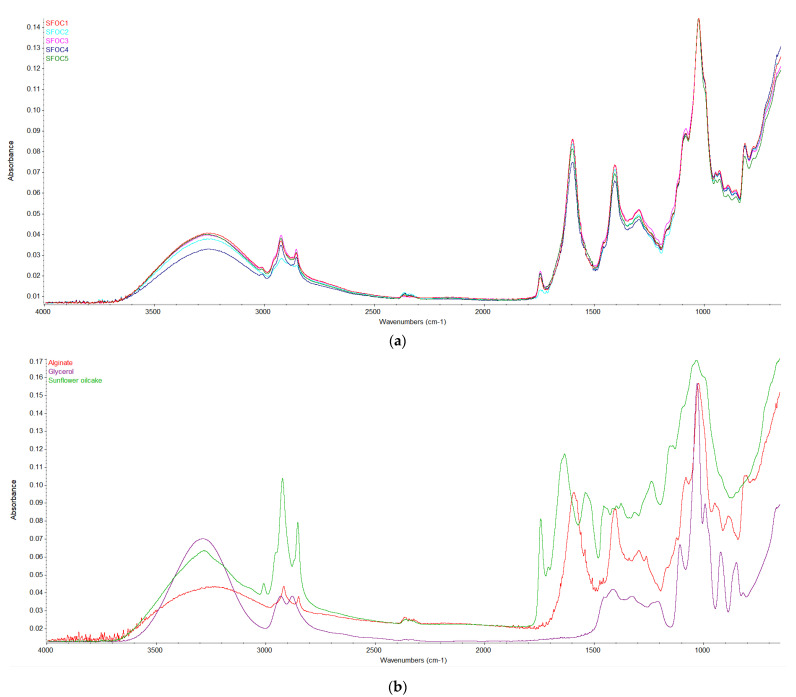
FT-IR spectra of the samples: (**a**) SFOC1, SFOC2, SFOC3, SFOC4, SFOC5 and control, (**b**) sunflower oilcake, alginate and glycerol.

**Figure 4 membranes-12-00789-f004:**
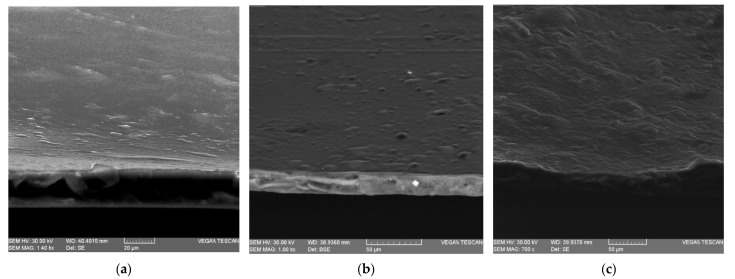
SEM images of the surface and cross-section of the control (**a**), SFOC1 (**b**) and SFOC5 (**c**).

**Figure 5 membranes-12-00789-f005:**
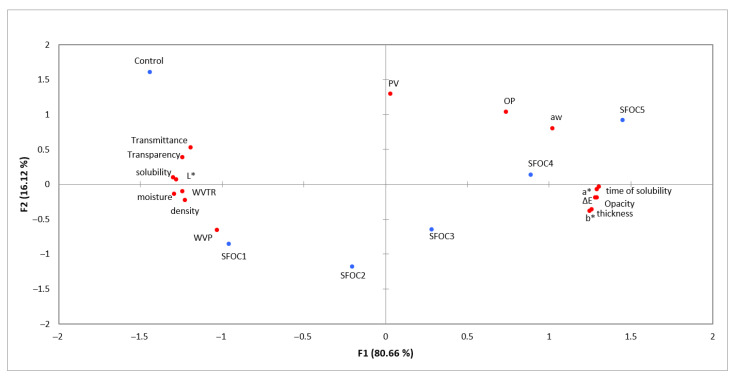
Principal component analysis bi-plot: distribution of the water-affinity, optical and barrier proprieties, density and thickness parameters, and samples.

**Table 1 membranes-12-00789-t001:** Edible-film formulations.

Sample	Sodium Alginate, g	Glycerol, g	Sunflower Oilcake, g	Water, mL
Control	1.00	0.50	0	100
SFOC1	1.00	0.50	0.1	100
SFOC2	1.00	0.50	0.2	100
SFOC3	1.00	0.50	0.3	100
SFOC4	1.00	0.50	0.4	100
SFOC5	1.00	0.50	0.5	100

SFOC1: film with 0.1 g of SFOC, SFOC2: film with 0.2 g of SFOC, SFOC3: film with 0.3 g of SFOC, SFOC4: film with 0.4 g of SFOC, SFOC5: film with 0.5 g of SFOC, Control: sodium alginate film.

**Table 2 membranes-12-00789-t002:** Study of mycotoxin incidence in SFOC.

Sample	Limit of Detection, µg/Kg	Limit of Quantification, µg/Kg	Results, µg/Kg	Maximum Limit *, µg/Kg
Zearalenone	10	15	35.22 ± 3.96	2000
Ochratoxin A	0.5	1.5	8.38 ± 1.36	50
Aflatoxin B_1_	0.3	0.7	<LOQ	10
Deoxynivalenol	0.011	0.042	<LOD	0.9

* according to 2006/576/EC.

**Table 3 membranes-12-00789-t003:** Water affinity of the membranes.

Sample	a_w_	Moisture Content, %	Solubility, %	Time of Solubility, min
Control	0.32 ± 0.01 ^c^	19.07 ± 0.78 ^a^	96.32 ± 0.11 ^a^	1.14 ± 0.08 ^a^
SFOC1	0.27 ± 0.01 ^d^	18.73 ± 0.23 ^a^	95.13 ± 0.39 ^a^	1.39 ± 0.11 ^b^
SFOC2	0.29 ± 0.01 ^d^	17.28 ± 0.80 ^b^	89.25 ± 0.69 ^a,b^	2.30 ± 0.09 ^c^
SFOC3	0.32 ± 0.01 ^c^	15.34 ± 0.76 ^c^	86.11 ± 0.84 ^a,b^	2.42 ± 0.08 ^c^
SFOC4	0.35 ± 0.01 ^b^	14.35 ± 0.60 ^d^	83.68 ± 4.71 ^b^	3.04 ± 0.04 ^d^
SFOC5	0.40 ± 0.03 ^a^	13.07 ± 0.42 ^e^	82.79 ± 2.87 ^b^	3.19 ± 0.16 ^e^

Different superscript letters after the values indicated differences statistically significant at *p* < 0.05%.

**Table 4 membranes-12-00789-t004:** Barrier properties and peroxide values of the tested membranes.

Sample	WVP, g × mm/KPa × h × m^2^	WVTR, g/h	PV, meq O_2_/kg	OP, g × mm × m^−2^ × day^−1^
Control	1.66 × 10^−4^ ± 5.00 × 10^−7 c^	13.55 ± 0.05 ^c^	5.50 ± 0.24 ^a^	0.027 ± 0.001 ^d^
SFOC1	1.98 × 10^−4^ ± 1.02 × 10^−5 d^	13.78 ± 0.74 ^c^	2.24 ± 0.14 ^d^	0.017 ± 0.000 ^d^
SFOC2	1.95 × 10^−4^ ± 2.45 × 10^−6 d^	11.56 ± 0.18 ^b^	2.24 ± 0.14 ^d^	0.018 ± 0.001 ^d^
SFOC3	1.37 × 10^−4^ ± 1.00 × 10^−6 b^	7.94 ± 0.05 ^a^	2.33 ± 0.00 ^d^	0.019 ± 0.001 ^c^
SFOC4	1.30 × 10^−4^ ± 1.00 × 10^−6 b^	7.34 ± 0.22 ^a^	3.50 ± 0.24 ^c^	0.030 ± 0.001 ^b^
SFOC5	1.13 × 10^−4^ ± 1.30 × 10^−6 a^	6.95 ± 0.93 ^a^	4.83 ± 0.24 ^b^	0.034 ± 0.000 ^a^

Different superscript letters (a,b,c and d) are significantly different (*p* < 0.05%) according to Turkey’s post hoc.

**Table 5 membranes-12-00789-t005:** Thickness, density, tensile strength and hardness values of the control and SFOC films.

Sample	Thickness, Mm	Density, g/cm^3^	TS, MPa	Hardness, N
Control	0.029 ± 0.01 ^a^	1.53 ± 0.03 ^a^	27.11 ± 1.97 ^a^	25.11 ± 0.19 ^a^
SFOC1	0.034 ± 0.002 ^b^	1.45 ± 0.03 ^a,b^	22.15 ± 1.36 ^b^	22.26 ± 0.74 ^b^
SFOC2	0.040 ± 0.002 ^c^	1.39 ± 0.01 ^b^	18.24 ± 0.22 ^c^	21.89 ± 0.26 ^c^
SFOC3	0.041 ± 0.003 ^c^	1.38 ± 0.02 ^b^	15.73 ± 0.28 ^d^	18.39 ± 1.69 ^d^
SFOC4	0.043 ± 0.003 ^c,d^	1.25 ± 0.04 ^c^	12.17 ± 0.39 ^e^	15.70 ± 0.50 ^e^
SFOC5	0.044 ± 0.003 ^d^	1.04 ± 0.01 ^d^	8.66 ± 0.77 ^f^	12.52 ± 0.50 ^f^

When followed by different superscript letters (a, b, c, d, e, f) they are statistically different at 95% confidence level.

**Table 6 membranes-12-00789-t006:** Optical properties of the tested films.

Sample	L*	a*	b*	ΔE	Opacity, UA/mm	Transparency, %/mm	Transmittance, %
Control	92.97 ± 0.33 ^e^	−5.53 ± 0.03 ^a^	10.31 ± 0.31 ^a^	1.74 ± 0.02 ^d^	6.50 ± 0.09 ^f^	65.90 ± 0.01 ^a^	81.52 ± 0.07 ^a^
SFOC1	87.04 ± 2.01 ^d^	−4.66 ± 0.37 ^b^	15.61 ± 1.31 ^b^	9.72 ± 0.55 ^c^	12.29 ± 0.03 ^e^	48.50 ± 0.02 ^b^	44.55 ± 0.06 ^b^
SFOC2	83.74 ± 1.68 ^c^	−4.11 ± 0.36 ^b,c^	18.95 ± 1.56 ^c^	14.47 ± 0.99 ^b^	17.30 ± 0.03 ^d^	34.50 ± 0.02 ^c^	23.98 ± 0.05 ^d^
SFOC3	81.01 ± 1.84 ^c^	−3.53 ± 0.34 ^c^	19.79 ± 1.42 ^cd^	17.10 ± 0.63 ^b^	17.95 ± 0.02 ^c^	33.92 ± 0.06 ^c^	24.54 ± 0.06 ^c^
SFOC4	76.78 ± 3.25 ^b^	−2.71 ± 0.74 ^d^	22.00 ± 2.44 ^d,e^	21.88 ± 0.07 ^a^	20.65 ± 0.02 ^b^	29.19 ± 1.25 ^d^	13.77 ± 0.50 ^e^
SFOC5	73.10 ± 0.73 ^a^	−2.10 ± 0.17 ^d^	23.46 ± 0.17 ^e^	25.77 ± 0.70 ^a^	26.06 ± 0.01 ^a^	20.90 ± 0.17 ^e^	8.38 ± 0.02 ^f^

Values followed by different superscript letters (a, b, c, d, e, f) are statistically different at 95% confidence level.

## Data Availability

Not applicable.

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
