# Peer review of "Sunflower Oilcake as a Potential Source for the Development of Edible Membranes"

_membranes, 2022, doi:10.3390/membranes12080789_

Round 1
Reviewer 1 Report
This is a revision of the manuscript „Superior Valorification of By-Products Resulting from Cold Pressing of Sunflower Seeds“. I commend the selection of the topic of the manuscript and the contribution to the research area, as well as the comprehensiveness of the applied methodology. I have several objections with the aim of improving the quality of work:
1. section Materials and methods: why did you use sodium alginate as basic solution where you added SFOC? Relate this fact with some literature data or if it is your original experimental design add info why you didnt use regular, for example, water solution? If your intention was to make composite films than state it when labeling samples. For example: it is confusing when you name Table 5 as „Thickness values of the alginate and alginate/SFOC films“ and in the table list samples as „control“ „SFOC1“, „SFOC2“ ... Please uniform this through entire manuscript – name samples as „alginate/SFOC1“, „alginate/SFOC2“...
2. section 2.2.: in text you referred to 1g of sodium alginate and 0.5g of glycerol but in Table 1 you referred to 1.02 g and 0.51g of sodium alginate and glycerol, respectively? Not clear.
3. section 2.3.: please change first sentence to „The safety of oilcake was determined by the following analysis: water activity, mineral content and mycotoxin incidence“.
4. please add section „2.4. Film characterization“ and all further section change to „2.4.1. Affinity to water“, „2.4.2. Barrier properties“, „2.4.3. Oxygen Permeability“, and so on.
5. When commenting on the results, especially oxygen permeability, oil permeability, color analysis, FTIR, morphology, antimicrobial properties, please refer to the literature references of other authors who examined alginate films, films based on sunflower cake or related biopolymer materials. This shortcoming in the presentation of results is the biggest shortcoming of the manuscript.
6. Correct the English language throughout the manuscript.
Considering that all remarks are correctable I suggest paper to be accepted after minor revision.
Author Response
Dear Editor,
We would like to thank the referee for the close reading and for the proper suggestions. We hope that we provided all the answers to the reviewers’ comments. Thank you very much for taking into account to publish our manuscript entitled "Sunflower oilcake as a potential source for the development of edible membranes" (Ms. ID
membranes-1852850)
The present version of the paper has been revised according to the reviewers and editor suggestions. For this purpose, Track Changes function has been used. All the modifications are marked with red color.
Also, we attach the response to the editor and reviewers.
We look forward to hearing from you soon. Sincerely yours,Ancuța PETRARU & Sonia AMARIEI,
R1 Comments and Suggestions for Authors
Reviewer: section Materials and methods: why did you use sodium alginate as basic solution where you added SFOC? Relate this fact with some literature data or if it is your original experimental design add info why you didn’t use regular, for example, water solution? If your intention was to make composite films than state it when labeling samples. For example: it is confusing when you name Table 5 as „Thickness values of the alginate and alginate/SFOC films“ and in the table list samples as „control“ „SFOC1“, „SFOC2“ ... Please uniform this through entire manuscript – name samples as „alginate/SFOC1“, „alginate/SFOC2“
Response: First of all, we would like to thank the referee for the close reading and for all the given comments suitable for improving the manuscript. The sodium alginate was used because it is resistant to solvents, oil, are water soluble and have good barriers to oxygen, water vapor. The information regarding sodium alginate were introduced in the introduction. We modified the labeling of the sample in the table and entire documents, so they are better understood.
Reviewer: section 2.2.: in text you referred to 1g of sodium alginate and 0.5g of glycerol but in Table 1 you referred to 1.02 g and 0.51g of sodium alginate and glycerol, respectively? Not clear.
Response: We would like to thank to the referee for the close reading.
We have clarified the information regarding the amounts of sodium alginate and glycerol.
Reviewer: section 2.3.: please change first sentence to „The safety of oilcake was determined by the following analysis: water activity, mineral content and mycotoxin incidence“.
Response: We would like to thank to the referee for the suggestion. We added more information in support of the sentences previously mentioned. Also we modified the sentence according to the referee suggestion.
Reviewer: please add section „2.4. Film characterization“ and all further section change to „2.4.1. Affinity to water“, „2.4.2. Barrier properties“, „2.4.3. Oxygen Permeability“, and so on.
Response: We would like to thank to the referee for the close reading and for all the given comments suitable for improving the manuscript.
We added and changed the title and numbers of sections according to the referee suggestions
Reviewer: When commenting on the results, especially oxygen permeability, oil permeability, color analysis, FTIR, morphology, antimicrobial properties, please refer to the literature references of other authors who examined alginate films, films based on sunflower cake or related biopolymer materials. This shortcoming in the presentation of results is the biggest shortcoming of the manuscript.
Response: We would like to thank to the referee for his remarks.
We added some references in the results section. Reviewer: Correct the English language throughout the manuscript.
Response: We would like to thank the referee for the close reading and all the remarks suitable for improving the manuscript. We correct the English throughout the manuscript.
Date Sent:
08.08.2022

Reviewer 2 Report
The article by Petraru and Amariei deals with the valorization of sunflower oilcake flour (SFOC), a by-product of the cold extraction of sunflower oil, as an alternative ingredient for the production of edible films based on alginate (not really mentioned in the title/abstract/introduction).
Different concentrations of SFOC (Characterized in previous work) were incorporated into the films and physico-chemical characterization was carried out.
The work is interesting, but sometimes concepts might be reinforced or a better explanation would be needed to enhance the results. Also, it needs an english revision. And check the dot, parenthesis and so on (the manuscript has no line number so it is difficult to write where the typo errors are)
Here some suggestion/questions
Abstract
I suggest removing the abbreviations for WVP and WVTR because you haven’t defined them yet.
“Moreover, the films with different SFOC levels were opaque, more colorful with tones ranging from light greenish/yellowish to dark. Films presented, also good protection against UV radiation.” In my opinion this sentence could be written in a better way
Introduction
I think the authors can improve this section especially the final part.
Table 1 . Please replace G with g
Par. 2.4
“where W0 is the mass of the film before drying (g) and W1 is the mass of the film after
drying (g), aw of the films was determined according to the methods described previously.”
Please after ‘drying (g)’ use a dot to finish the sentence.
Par 2.5.2
0.01n change with N
Par 2.10
35° add C
Results
3.1
SFC replace with SFOC
3.2
SFC replace with SFOC
Correct “wich” and “doen”
“The solubility of the films was high. When oilcakes was used solubility decrease, the lowest was found for SFOC5” please reformulate
Table 3- why did you not report the significance/statistic for solubility?
“WVP of films ranged between 1.13 × 10−4 and 1.98 × 10−4 g*mm/KPa*h*m2 and was significant (p < 0.05) affected by the amount of SFOC added” please correct
3.3.2 oxygen
“The PV results obtained for uncovered oil was 7.67 meq O2/Kg, which was higher that the PV values obtained for the membrane covered oil. The results indicates that SFO can block oxygen transmission.”
Please reformulate (why now you use SFO?)
“The increasing is related to the high content in fats on the oilcakes” please reformulate and correct
3.3.2 oil (change the number)
“After adding SFOC the values increased from 0.017 to 0.034 g mm m−1 day−1. No significant difference were found between the control sample and the membranes with 0.1g and 0.2 g SFOC. In contrary, significant difference were found when the addition of flour increased (0.4g-0.5g). In conclusion the films based on SFOC could be used for the packaging of oil-rich food.”
Why do you say that SFOC-based films could be used for the packaging of foods rich in oil if increasing its content increases oil permeability?
Is there any reference for the oil permeability values of packaging from the literature?
3.4
you always use the p value of statistic lower than 0.05 and here you introduce 95%. Why?
“This may be associated with the increasing in solid contents, the various difference in structure and chemical composition (high fiber content of SFOC) and the hydrophobicity character of the film constituents.” Please correct
3.5.2
“ Protection against UV radiation is also important for the packaging material because they can cause a deterioration of the packaging material “ please reformulate
3.6
“SFOC membranes are shown in Figure.” Number of the figure missing
Figure 3 is it impossible to try to have more spectra on the same graph? At least the one of the films?
3.7
“easy to handle and resistance when manipulated by hand” correct
“In Figure 4 are presented the SEM micrograph of the alginate and alginate/SFOC composite films The control sample presented a smooth and uniform structure” correct
3.8
Does the alginate alone have antimicrobial activity? (check other study ?) Could you talk about an antimicrobial effect?
Table 7 are the number in the table the CFU? How do you explain the result for enterococcus and yeast and molds for the two different formulation sfoc1 and 5?
Remeber to use italic for microorganisms.
Conclusions
I suggest trying to make them more concrete and avoid some repetition
In my opinion the manuscript needs major revision before being published.
Author Response
Dear Editor,
We would like to thank the referee for the close reading and for the proper suggestions. We hope that we provided all the answers to the reviewers’ comments. Thank you very much for taking into account to publish our manuscript entitled "Sunflower oilcake as a potential source for the development of edible membranes" (Ms. ID
membranes-1852850)
The present version of the paper has been revised according to the reviewers and editor suggestions. For this purpose, Track Changes function has been used. All the modifications are marked with red color.
Also, we attach the response to the editor and reviewers.
We look forward to hearing from you soon. Sincerely yours,Ancuța PETRARU & Sonia AMARIEI,
R2 Comments and Suggestions for Authors
Reviewer: I suggest removing the abbreviations for WVP and WVTR because you haven’t defined them yet.
Response: First of all, we would like to thank the referee for the given remark suitable for improving the manuscript. We removed the abbreviations and substituted with their definition
Reviewer: “Moreover, the films with different SFOC levels were opaque, more colorful with tones ranging from light greenish/yellowish to dark. Films presented, also good protection against UV radiation.” In my opinion this sentence could be written in a better way
Response: We rewritten the sentence to be better understood Reviewer: Introduction- I think the authors can improve this section especially the final part.
Response: We would like to thank to the referee for the close reading. We improved the introduction section by adding more information regarding the work of other researchers.
Reviewer: Table 1 . Please replace G with g
Response: We replaced according to the referee suggestion
Reviewer: “where W0 is the mass of the film before drying (g) and W1 is the mass of the film after drying (g), aw of the films was determined according to the methods described previously.”
Please after ‘drying (g)’ use a dot to finish the sentence.
Response: We would like to thank the reviewer for the close reading and for all the given comments suitable for improving the manuscript. We added the dot
Reviewer: 0.01n change with N
Response: We change the according to the referee suggestion
Reviewer: 35° add C
Response: We would like to thank the referee according to the referee suggestion. We added
Reviewer: SFC replace with SFOC
Response: We replaced the word according to the referee suggestion
Reviewer: “The solubility of the films was high. When oilcakes was used solubility decrease, the lowest was found for SFOC5” please reformulate
Response: We would like to thank the referee for the suggestion. We reformulate the sentence.
Reviewer: Table 3- why did you not report the significance/statistic for solubility?
Response: We added the superscript regarding the statistic analysis
Reviewer: “WVP of films ranged between 1.13 × 10−4 and 1.98 × 10−4 g*mm/KPa*h*m2 and was significant (p < 0.05) affected by the amount of SFOC added” please correct
Response: We would like to thank the referee for the close reading. We reformulate the sentence
Reviewer: “The PV results obtained for uncovered oil was 7.67 meq O2/Kg, which was higher that the PV values obtained for the membrane covered oil. The results indicates that SFO can block oxygen transmission.”Please reformulate (why now you use SFO?)
Response: We would like to thank the referee for the suggestion. We reformulate the sentence to be better understood
Reviewer: “The increasing is related to the high content in fats on the oilcakes” please reformulate and correct
Response: We would like to thank the referee for the suggestion. We reformulate the sentence to be better understood
Reviewer: “After adding SFOC the values increased from 0.017 to 0.034 g mm m−1 day−1. No significant difference were found between the control sample and the membranes with 0.1g and 0.2 g SFOC. In contrary, significant difference were found when the addition of flour increased (0.4g-0.5g). In conclusion the films based on SFOC could be used for the packaging of oil-rich food.” Why do you say that SFOC-based films could be used for the packaging of foods rich in oil if increasing its content increases oil permeability?
Is there any reference for the oil permeability values of packaging from the literature?
Response: We would like to thank the referee for the suggestion. We added a reference for the oil permeability. The results obtained in the previous study compared with ours were much higher. The results being lower it is possible to say that are a promising material for the packaging of oil-rich food.
Reviewer: you always use the p value of statistic lower than 0.05 and here you introduce 95%. Why?
Response: We referred to the confidence interval
Reviewer: “This may be associated with the increasing in solid contents, the various difference in structure and chemical composition (high fiber content of SFOC) and the hydrophobicity character of the film constituents.” Please correct
Response: We would like to thank the referee for his kind comment. We correct the sentence according to the referee suggestion
Reviewer: “ Protection against UV radiation is also important for the packaging material because they can cause a deterioration of the packaging material “ please reformulate
Response: We would like to thank the referee for the suggestion
Reviewer: “SFOC membranes are shown in Figure.” Number of the figure missing
Response: We added the number of the figure according to the referee suggestion
Reviewer: Figure 3 is it impossible to try to have more spectra on the same graph? At least the one of the films?
Response: We changed the imagines regarding the FT-IR analysis. We condensed the spectra of the developed films in one imagine and the spectra regarding the main ingredients in another imagine.
Reviewer: “easy to handle and resistance when manipulated by hand” correct
Response: We correct the sentence to be better understood
Reviewer: Does the alginate alone have antimicrobial activity? (check other study ?) Could you talk about an antimicrobial effect?
Response: We would like to thank the referee for his kind suggestion suitable for improving the manuscript. The sodium alginate forms a general protection that increase the resistance against microbial agents. We added also some references.
Reviewer: Table 7 are the number in the table the CFU? How do you explain the result for enterococcus and yeast and molds for the two different formulation sfoc1 and 5?
Remeber to use italic for microorganisms.
Response: We would like to thank the referee for the suggestion. We used italics for the microorganism. We added the measure unity in table 7. Also, we modified the results obtained for enterococcus, yeast and mold for SFOC1 they were wrong. Were added also the explication regarding the results obtained for SFOC5.
Reviewer: Conclusion. I suggest trying to make them more concrete and avoid some repetition
Response: We would like to thank the referee for the suggestion. We reformulate the conclusions to be more concrete.
Date Sent:
08.08.2022

Round 2
Reviewer 2 Report
The authors responded to the suggestions that were made. the work presents preliminary results that could be interesting for future research.
Here a series of points to check/clarify:
Why are you now talking about edible membrane in the title and in the test?
Line 32 controls the repetition of "obtain”
Line 36-37 "alginates ... is" please edit
Line 47 "considered for comercialized" please change
Line 78 'with the increase of the amount' please add the subject
Line 81 - what were the components of these high performance films? Please add in the sentence
Line85- In my opinion the aim of the work could still be improved
check Cfu / cfu (always write the same way)
Please double check the english in the manuscript
In my opinion the manuscript can be accepted after a little revision
Author Response
Dear Editor,
We would like to thank the referee for the close reading and for the proper suggestions. We hope that we provided all the answers to the reviewers’ comments. Thank you very much for taking into account to publish our manuscript entitled "Sunflower oilcake as a potential source for the development of edible membranes" (Ms. ID
membranes-1852850)
The present version of the paper has been revised according to the reviewers and editor suggestions. For this purpose, Track Changes function has been used. All the modifications are marked with red color.
Also, we attach the response to the editor and reviewers.
We look forward to hearing from you soon. Sincerely yours,Ancuța PETRARU & Sonia AMARIEI,
R2 Comments and Suggestions for Authors
Reviewer: Why are you now talking about edible membrane in the title and in the test?
Response: First of all, we would like to thank the referee for the given remark suitable for improving the manuscript. The title was modified according the editor suggestion. The initial idea was that the developed films should be edible.
Reviewer: Line 32 controls the repetition of "obtain”
Response: We rewritten the sentence excluding the repetition of the word "obtain”.
Reviewer: Line 36-37 "alginates ... is" please edit
Response: We would like to thank to the referee for the close reading. We corrected the sentence.
Reviewer: Line 47 "considered for comercialized" please change
Response: We changed the word according to the referee suggestion
Reviewer: Line 78 'with the increase of the amount' please add the subject
Response: We would like to thank the reviewer for the close reading and for all the given comments suitable for improving the manuscript. We added the subject
Reviewer: Line 81 - what were the components of these high performance films? Please add in the sentence
Response: We change and added the components according to the referee suggestion
Reviewer: Line85- In my opinion the aim of the work could still be improved
Response: We would like to thank the referee for the suggestion. We improved the aim of the work to be better understood by the readers.
Reviewer: check Cfu / cfu (always write the same way)
Response: We corrected the word throughout the document according to the referee suggestion
Reviewer: Please double check the english in the manuscript
Response: We would like to thank the referee for the close reading and all the suitable comments suitable to improve the manuscript. We checked the english in the manuscript.
Date Sent:
13.08.2022
